# Use of Electronic Cigarettes in European Populations: A Narrative Review

**DOI:** 10.3390/ijerph17061971

**Published:** 2020-03-17

**Authors:** A. Kapan, S. Stefanac, I. Sandner, S. Haider, I. Grabovac, T.E. Dorner

**Affiliations:** 1Department of Social and Preventive Medicine, Centre of Public Health, Medical University of Vienna, Vienna 1090, Austria; sinisa.stefanac@meduniwien.ac.at (S.S.); n1140406@students.meduniwien.ac.at (I.S.); sandra.a.haider@meduniwien.ac.at (S.H.); igor.grabovac@meduniwien.ac.at (I.G.); thomas.dorner@meduniwien.ac.at (T.E.D.); 2Institute of Outcomes Research, Centre for Medical Statistics, Informatics and Intelligent Systems, Medical University of Vienna, Vienna 1090, Austria

**Keywords:** prevalence, e-cigarettes, current and ever-use, trend

## Abstract

The increasing popularity of electronic cigarettes in past decades has aroused public health concern. This study aims to review the literature on the prevalence of e-cigarette use among the general adult and young populations in Europe. We searched Medline and Google Scholar from September 2019, and included “prevalence of e-cigarettes”, “electronic cigarettes” or “e-cigarettes”, and “electronic nicotine delivery system” or “vaping”. The prevalence of current e-cigarette use ranged from 0.2% to 27%, ever-use ranged from 5.5% to 56.6% and daily use ranged from 1% to 2.9%. Current smokers of conventional cigarettes showed the highest prevalence for the use of e-cigarettes, ranging from 20.4% to 83.1%, followed by ex-smokers, with ranges from 7% to 15%. The following socio-demographic factors were associated with a higher chance of using e-cigarettes: male sex and younger age groups; results for economic status were inconclusive. In European countries, there is a higher prevalence of e-cigarette use among males, adolescents and young adults, smokers of conventional cigarettes, and former smokers.

## 1. Introduction

Electronic nicotine delivery systems (ENDS) are marketed under a variety of names, most commonly-referred to as “electronic cigarettes (e-cigarettes),” but also as “e-cigs”, “vapes”, “vape pens”, and “mods”. These different types of electronic nicotine delivery systems are designed to be either less harmful than regular cigarettes or used as nicotine replacement therapy (NRT) [1]. The electronic nicotine device generally consists of a power source, usually a battery, and a heating element that creates an aerosol that is inhaled by the user after the e-liquid (the solution inside a device) has been heated to a temperature of above 350 °C [2,3].

There is currently active debate about benefits and harms of e-cigarettes at the individual and population level. First, there is uncertainty and debate about the degree to which e-cigarettes help existing smokers to quit. The latest Cochrane Database Systematic Review and meta-analysis found that participants using nicotine-containing e-cigarettes are approximately two-and-a-half times more likely to have abstained from smoking for at least 6 months, compared to those using placebo e-cigarettes. However, the authors have also noted an overall lack of studies and found that the available studies were of low quality and had generally small sample sizes [4]. A limited number of randomized clinical trials evaluating e-cigarette use for smoking cessation have been published [5,6,7], and the results are conflicting. The latest study mentioned above showed that e-cigarettes were more effective than NRT [7]. However, it is important to note that this study also differs from an earlier trial [5] in that participants demonstrated motivation to quit a priori. Moreover, given the constant emergence of new studies, our as-of-yet unpublished meta-analysis found that e-cigarettes with nicotine showed a tendency to be effective in smoking cessation, as compared to placebo e-cigarettes without nicotine. However, the level of evidence was moderate to low, and the analysis results were not significant [8].

Further, little common ground is found among health organizations regarding the question of e-cigarette use in smoking cessation. For example, Public Health England supports e-cigarette use for smoking cessation [9], whereas US health agencies concluded there is insufficient evidence to recommend e-cigarettes use for cessation [10,11]. The National Academies of Sciences raised concerns due to unanswered questions regarding long-term health effects in users, as reports suggest that e-cigarettes may damage various organ systems [12]. Moreover, the Centers for Disease Control and Prevention (CDC) has declared an ongoing epidemic of e-cigarette or vaping use associated with lung injury (EVALI) throughout the United States [13,14]. As of 7 January 2020, vaping-related lung injuries have caused 57 confirmed deaths in 27 states and the District of Columbia. Data show that vitamin E acetate, an additive in some tetrahydrocannabinol- (THC) containing e-cigarettes, is strongly linked to the EVALI outbreak, while vitamin E acetate has not been found in the lung fluid of people that do not have EVALI. For this reason, the CDC recommends that people not use e-cigarette products that contain THC [15].

While vaping e-cigarettes may represent a form of harm reduction for adult smokers, there are concerns about potential harm for adolescents, including the risk that ENDS use may act as a gateway to smoking cigarettes among young people [16]. As a meta-analysis from 2016 shows, among never-smoking adolescents and young adults, e-cigarette use was associated with increased smoking intention, as compared to peers who did not use e-cigarettes [17].

In order to estimate the impact of e-cigarettes (both positive and negative), it is important to understand the prevalence of e-cigarette use in the general population. Several studies on the prevalence of e-cigarette use have been already published [18,19,20,21]. However, these are mostly based in the United States of America, with few focusing on the European continent. This may be problematic as cultural and public health differences may prohibit generalizations of US-based results on the European context. Furthermore, with the growing availability of e-cigarettes, an update of the prevalence is important [22]. In light of such a rise in scientific interest and the number of publications, we aim to carry out a narrative review of the available literature on e-cigarette use in European population.

## 2. Materials and Methods 

In order to perform a narrative review of available literature on prevalence of e-cigarette use, we conducted a literature review in September 2019 in Medline and Google Scholar using the terms “prevalence of electronic cigarettes use”, “e-cigarettes use”, “electronic nicotine delivery system”, “vaping”, and “frequency of e-cigarette use”. The reference lists of articles that were found were additionally screened for potential articles.

The retrieved articles were screened for content and were selected if (1) they were written in English, if (2) the data provided population-based estimates of e-cigarette use in adults and/or adolescents from one-or more countries of the WHO European region, and (3) if the article was published in a peer-reviewed journal (4) between 2011 and 2019. Full text articles were obtained only if the abstracts included data about the prevalence of e-cigarettes with a cross-sectional or a longitudinal design. Collectively, we summarized the results describing the themes relevant to prevalence and factors of e-cigarette use. If more studies on the same prevalence data were given, data were shown in ranges (minimum to maximum as percentage); otherwise the prevalence values were shown as single percentages.

## 3. Results

Overall, 22 studies were included in the review (Table 1). Of these, 4 of them included data from multiple countries [22,23,24,25], and 18 studies presented data from single countries [26,27,28,29,30,31,32,33,34,35,36,37,38,39,40,41,42,43]. All included studies were either cross-sectional studies, or cross-sectional baseline findings of longitudinal studies. The sample sizes ranged from 726 to 27901 subjects.

### 3.1. Prevalence of Using E-Cigarettes in the General Population

In Table 1, the findings regarding the use of e-cigarettes in the general population are summarized. The prevalence of current e-cigarette smokers (the definition of current use of e-cigarettes varied among the surveys, from “vaped at least one e-cigarette in the last 30 days” to “at least one e-cigarette per day at the moment of the survey”) ranged from 0.2% to 27%, those who reported ever trying ranged from 5.5% to 56.6%, and between 1% and 2.9% were found to be daily e-cigarette users. There were differences in age among participants who had tried e-cigarettes. The highest prevalence was found among those aged 10–24 years (5.5% to 56.6%), followed by those aged 25–39 (13.7% to 25%), 40–65 (5% to 6.7%), and those aged ≥ 65 years (1.3% to 1.6%). For example, in a sample of 5385 Serbians, there were about 3 times more current e-cigarette users among 25–44-year-olds than among 55–64-year-olds (3% vs. 1.1%) [28]. It seems to point to a trend that with increasing age, the use of e-cigarettes decreases.

### 3.2. Spacial Differences in Using E-Cigarettes within the WHO European Region

European regions showed a varying picture; southern regions showed similarities, with the reported prevalence of ever-use in Italy and Spain ranging from 5.6% to 6.5%. In northern regions, however, the prevalence ranged from 12% to 17.4% in Finland, to up to 26% in Sweden. We also observed differences between western and eastern European regions. Low prevalence was mostly found in western European countries, with the following prevalence rates: France (17.9% to 54%) Netherlands (29.4%), Ireland (24%), Germany (11.8%), England (7.4%), Wales (5.8%), and the lowest prevalence in Switzerland (4.9%). In comparison, highest prevalence was reported among eastern European countries, with highest being in Lithuania (56.65%), followed by Poland (20.9% to 45%), Belarus (42.7%), Slovakia (34.4%), Russia (33.4%), and with considerably lower prevalence of ever-use being reported in Serbia (less than 10%). Thus in general, the results indicate higher prevalence among eastern WHO European region countries.

### 3.3. Gender and Ethnic Difference 

Men showed higher prevalence rates of e-cigarettes use than women. In included studies, men showed up to 5 times higher prevalence of e-cigarette use than women. Further, daily use was more common among men (1.5%) than among women (0.9%). In the study of Jawad et al. (2015), differences among ethnic groups are reported. When comparing Caucasians with other ethnicities, studies report more use among other ethnic groups (14.9% versus 5.6%; adjusted OR 1.76, 95% CI 1.13 to 2.73) [32].

### 3.4. Socio-Economic Differences 

Studies from Italy [30] and Spain [33] show that participants with secondary school education were more likely to have ever used e-cigarettes than those who reported their educational level as being “low” or “high”. Current use among participants with differing educational levels ranged from 0.5% in those reporting high educational level, to 1% in those with low, and 1.6% in those with secondary level. Different results are shown in an English study reporting differences in e-cigarette use, in which smokers with a higher social grade (based on occupation) also showed higher e-cigarette use than those with a lower social grade [31]. The use of e-cigarettes in long-term ex-smokers increased over time among all groups, and was far more common in groups with lower socio-economic status. In the analyses, respondents were stratified by socio-economic status using the National Readership Survey classification system for social grade based on the occupation of the main income earner, which has useful discriminatory power as a target group indicator [31]. In addition, a study from Poland shows that participants whose parents had a primary-level education indicated current e-cigarette use more frequently than those whose parents had a tertiary-education level [35].

In terms of employment status and household income, the odds of being an ever e-cigarette user increased with lower income and unemployment (OR = 2.9), as compared to those with employment and higher income. Among respondents who had at least tried e-cigarettes, socio-economic differences showed the highest prevalence among the unemployed (25%), manual workers (29%), students (19%), and the self-employed (18%), followed by other white-collar workers (16%), managers (12%), housewives (8%), and retired persons (6%) [22].

### 3.5. E-Cigarette Use and Smoking Status

Current smokers of conventional cigarettes showed the highest prevalence for ever-use of e-cigarettes, ranging from 20.4% to 83.1%. This is followed by ex-smokers, with prevalence rates ranging from 7% to 15%. Using e-cigarettes was rare among non-smokers, with a prevalence ranging from 2.3% to 5.6% for ever-use. For example, a cross-sectional survey of a French population aged 15–75 years old showed that more than 98% of current e-cigarette users were, or had been, conventional cigarette smokers [27].

The concern that young people who use e-cigarettes may be more likely to smoke cigarettes in the future [44] can be partly confirmed by the study of Treur et al. (2018) [38]. Adolescents who ever used an e-cigarette with nicotine were 11.90 more likely (95% CI 3.36 to 42.11) to smoke a conventional cigarette 6 months later than those who never used an e-cigarette with nicotine. On the contrary, the odds of smoking a conventional cigarette 6 months after smoking an e-cigarette without nicotine were 5.36 (95% CI 2.73 to 10.52) and 5.36 (95% CI 2.78 to 10.31) for water pipe. An additional study shows that the percentage of e-cigarette ever-users and reported current smokers increased from 6.9% among 10–11-year-olds to 39.2% among 15–16-year-olds. Current use of e-cigarettes was more likely among those who had previously smoked tobacco. Eighty percent of current e-cigarette users reported having also smoked cigarettes, compared to 72.1% of young people who had used an e-cigarette a few times, and 43.2% of current e-cigarette users were not current smokers [36].

### 3.6. Type of E-Cigarette Used (Nicotine or Non-Nicotin) 

One study reports that 77% of the current e-cigarette users always used nicotine-containing e-liquids, 14% sometimes, and 9% never used nicotine-containing e-liquids. Fifty percent of ever-users stated always using nicotine-containing e-liquids [29]. Another study shows that among current users, 95.5% used e-cigarettes with nicotine and the remaining 4.5% used e-cigarettes with vapor and flavors only [30]. In contrast in Spain, 62.5% of ever-users tended to use e-liquids with nicotine [33].

A different picture can be seen in the younger population. One study shows that 65.7% of e-cigarette ever-users tended to use nicotine e-liquids (among these, 2.9% were never-smokers), 23.5% used liquids without nicotine, and 10.9% did not know whether the liquid had contained nicotine or not [37]. In addition, adolescents in Sweden reported more use of e-cigarettes with nicotine (13%) compared to e-cigarettes without nicotine (10%) [42]. In a Dutch cohort however, the prevalence of ever-use of e-cigarettes with nicotine was 13.7% (11 to 17 years) and 12.3% (14 to 21 years), respectively, whereas the prevalence of e-cigarette use without nicotine was 29.4% (11 to 17 years), and 27.6% (14 to 21 years), respectively. In the group of current users, the mean number of times used in the past month was highest for e-cigarettes with nicotine, or 11.1 (SD = 14.5) in 11–17-year-olds and 9.3 (SD = 13.9) in 14–21-year-olds, compared to those using e-cigarettes without nicotine, 7.9 (SD = 12.0) and 4.8 (SD = 9.5), respectively [38].

### 3.7. Trends in Using E-cigarettes

Time trends in using e-cigarettes can be derived from the Eurobarometer, which was carried out in 2014 and 2017 with similar methods [22,23]. Data show that 1.5% (95% CI 1.2 to 1.8) of the adult population in the European Union in 2014 were currently e-cigarette users, compared to 1.8% (95% CI 1.5 to 2.1) in 2017, respectively. Additionally, the prevalence of e-cigarette ever-use increased from 2012 (7.2%) to 2017 (14.6%) [23].

### 3.8. Reasons for E-cigarette Use

The most frequent reasons for starting the use of e-cigarettes were to stop or reduce tobacco consumption (61%), because e-cigarettes were seen as less harmful (31%), had lower costs (25%), and that e-cigarette use is allowed in areas where regular tobacco smoking is not (15%); other reasons included different flavors (12%), that friends were also taking up e-cigarette smoking (11%), and that e-cigarettes were perceived as cool or attractive (6%) [22]. A study by Filippidis et al. (2017) of adults from 27 European countries shows that the main reason for using e-cigarettes among current e-cigarette users was that they believed e-cigarettes could help them quit smoking, and because they wanted to circumvent smoking bans [24].

A further study of people aged 14 years or over from Germany found that the main reasons for e-cigarette use in ever-users were “curiosity” (59%), followed by “quitting tobacco use or nicotine use” (29.1%), “complement to smoking” (7.8%), and “other reasons” including taste and lower price (2.1%). Current e-cigarette users most frequently named “quitting tobacco or nicotine use” (52%), followed by “complement to smoking” (25%), and “curiosity” (12.5%) as their reasons. Among smokers, “quitting tobacco or nicotine use” (46%), and among young people, “curiosity” (73%) were the main reasons for e-cigarette use [26].

The reasons for e-cigarette use found by Andler et al. (2016) in a survey of adults from France were addiction to nicotine (three quarters of e-cigarette users), the consideration of e-cigarettes being less harmful than conventional cigarettes (named by 60% of dual users and 80% of former smokers who vaped), e-cigarettes being less expensive (stated by 66% of dual users and 71% of vaping ex-smokers), and being permitted in places where conventional cigarettes are banned (reason for 28% of dual users and 20% of vaping ex-smokers). They also found that among dual users, 69.4% wanted to quit smoking conventional cigarettes, as compared to 54.2% among non-vaping smokers [27].

## 4. Discussion

The results of our review show that the European population’s lifetime-prevalence of using e-cigarettes is high, whereas prevalence of current daily smoking of e-cigarettes is quite low. In 2018, 3.2% of US adults reported current e-cigarette use [45], which is similar to our findings in the WHO European region. However, there are major differences in subpopulations in Europe. Our review shows much lower prevalence of e-cigarette use among older adults who have never smoked. However, the prevalence seems to be on the rise. In summary, daily e-cigarette use was much more common among smokers or former smokers [23]. There was evidence of variation in e-cigarette use by ethnicity and region. For example, the survey by Jawad et al. (2015) found that there was more e-cigarettes use among non-Caucasian ethnic minorities [32]. This is in line with a four-country survey (Canada, USA, United Kingdom, and Australia) from 2013 [46] that found that there was generally higher awareness of e-cigarette use among the Caucasian ethnicity compared with non-Caucasian ones. There are other disparities, such as spatial differences in e-cigarette use. People from eastern European countries used e-cigarettes more often than the European regional average. This may be due to differences in tobacco control policies and different accessibility to tobacco. For instance, Czech Republic, Slovakia, and Poland have weak implementation of smoke-free public rooms, especially in the hospitality industry [47]. Further, in poorer countries, people tend to smoke more; socioeconomic inequality is apparent in initiation: the risk that young people will start smoking is higher in less privileged groups [48]. Such disparity calls for possible policy interventions that can help accelerate the reduction of e-cigarette use in these areas.

Studies show that adolescents who ever used an e-cigarette with nicotine were more likely to smoke cigarettes in the future [38,44]. A large proportion of current e-cigarette users reported having also smoked cigarettes, but almost three-quarters of young people who had used an e-cigarette a few times, and almost half of current e-cigarette users, were not current smokers [36]. Further, among young adults, experimentation with e-cigarette use increased with advancing age, among daily smokers, best friends being smokers, and those whose siblings were smokers. [39,42]. Young males were also slightly more likely to experiment with e-cigarettes than females. The data suggest that peers may influence experimentation in young populations [37,39,40]. Another reason of e-cigarette experimentation may also be that sensation-seeking, or the need for new, different, or complex sensations and experiences-and the willingness to take risks to achieve them-is associated with adolescent substance use [49]. Further, studies shows that sweet flavors and smells are disproportionately appealing to youth, and are cited as a primary reason for use among this age group relative to adults [50]. On the basis of the above, regulation of flavor chemicals in e-cigarette products should be addressed, given that preferences for specific sweet flavors predicted e-cigarette use exclusively among youth. Overall, these results provide some support for the hypothesis that e-cigarettes act as a gateway to conventional cigarette smoking, though other explanations for the association are possible. The findings of high prevalence of use among adolescents and young adults suggest that e-cigarettes have the potential to expand the nicotine market in these age groups and may have the effect of renormalising smoking. Further monitoring and research to investigate these issues is required.

### Limitations & Directions for Future Research

One limitation is that only one author conducted the narrative review process. It is possible that another reviewer may have included additional information. However, our study is based on published evidence and offers an overview of the use of electronic cigarettes in European populations.

The findings are limited by the quality of the methods of the surveys in the included studies. For example, some questionnaires were not validated, thus the extent to which they capture true prevalence is unclear, as is the extent to which this affected the internal validity of the findings. Furthermore, wording of questions assessing e-cigarettes may have changed, potentially introducing misclassification bias. A lack of a unified definition of what constitutes “use” of an e-cigarette is also a challenge for the survey research, while measures should aim to also capture, for example, “current daily use” or other factors.

## 5. Conclusions

Overall, the results suggest that e-cigarettes are used predominantly by smokers and former smokers. There is a higher prevalence of e-cigarette use among males, adolescents, and young adults, as well as within populations of eastern European countries. For adolescents and young adults, additional research is recommended to identify whether e-cigarettes encourage or reduce uptake of smoking and support smoking cessation.

## Figures and Tables

**Table 1 ijerph-17-01971-t001:** Prevalence of e-cigarette use among the general population.

Citation	Data Source	Country	Sample Characteristics	Findings
**European Commission, Special Eurobarometer 2017** [22]	2017 Eurobarometer survey	28 Member States of the European Union	27901 respondents from different social and demographic groups	Respondents who have at least tried e-cigarettes: 15%, Respondents who have tried them once or twice but do not use them currently: 9% overall, Respondents who currently use e-cigarettes or similar electronic devices: 2%, 67% of current users used e-cigarettes daily, 20% reported weekly use, 7% monthly and 6% stated using e-cigarettes less than monthly; Respondents who are daily e-cigarette users: 1%, Respondents who used to use them but do not use them anymore: 4%, Current conventional smokers: - who currently use e-cigarettes or similar devices: 4% in EU28 - who used to use e-cigarettes but no longer do so: 10% in EU28 - who tried e-cigarettes once or twice: 23% in EU28, Ex-smokers: - who currently use e-cigarettes or similar devices: 4% in EU28 - who used to use e-cigarettes but no longer do so: 4% in EU28 - who tried e-cigarettes once or twice: 7% in EU28, Never-smokers: - who currently use e-cigarettes or similar devices: 0% in EU28 - who used to use e-cigarettes but no longer do so: 1% in EU28 - who tried e-cigarettes once or twice: 2% in EU28, Among respondents who have at least tried e-cigarettes socio-demographic differences show highest prevalence among: - men (17%) compared to women (12%) - those aged 15–24 (25%) followed by 25–39 (21%), 40–54 (15%) and 55+ (6%) - participants that were still studying (19%) followed by those who left full time education at the age of 16–19 (16%) age 20+ (14%) and age 15 and before (8%) - manual workers (29%), the unemployed (25%, students (19%), and the self-employed (18%) followed by other white collars (16%), managers (12%), household persons (8%) and the retired (6%) - those who stated having difficulties paying bills “most of the time” (23%) compared to those who stated having such problems “from time to time” (18%) or “never/almost never” (12%),
**Laverty et al. 2016** [23]	2014 and 2017 Adult Special Eurobarometer for Tobacco Survey	28 Member States of the European Union	2014: 27801 respondents 2017: 27901 respondents as representative samples of the population aged ≥ 15 years in each of the 28 EU member states and across the EU in terms of age, gender and area of residence.	Prevalence of ever e-cigarette use 2014: 11.6%; 2017: 14.6%, Statistically significant increases (from 2014 to 2017) in ever-use were found in 15 EU member states, highest in Belgium (OR = 3.45), Ever-use was more likely among: - men (OR = 1.25) than among women - younger people aged 15–24 (OR = 8.23) compared to aged 25–39 (OR = 3.71) and 40–54 years old (OR= 2.10), - people with more years in education (OR = 1.59 for those completing education at age ≥ 20 years compared with ≤ 15 years), - former (OR = 7.49) and current tobacco smokers (OR = 22.88) compared to never-smokers, Among current cigarette smokers, heavy smokers were more likely to have ever used e-cigarettes than light smokers (OR = 26.59), Current e-cigarette use: - 2014: highest prevalence was in France and the United Kingdom (3.6%) lowest prevalence in Malta (0%), - 2017: highest prevalence again in the United Kingdom (4.7%) and France (3.7%) and lowest in Bulgaria (0.2%), Overall current e-cigarette use 2014: 1.5%; 2017: 1.8%, 86.7% of current e-cigarette users reported current use, Among ever e-cigarette users younger people were less likely to have become ever current users (p for trend across age groups < 0.001), Ever-use (OR = 1.46) as well as current use (OR = 1.32) of e-cigarettes were more common in 2017 than in 2014,
**Filippidis et al. 2017** [24]	2012 and 2014 Adult Special Eurobarometer for Tobacco Survey	27 Member States of the European Union (excluding Croatia)	2012: 26751 respondents 2014: 26792 respondents as representative Samples of the population aged ≥15 years in each of the 27 EU member states (excluding Croatia) and across the EU in terms of age, gender and area of residence.	Prevalence of e-cigarette ever-use 2012: 7.2%; 2014: 11.6%, EU-wide coefficient of variation in e-cigarette ever-use: 42.1% in 2012 and 33.4% in 2014, Ever e-cigarette use in 2014 varied from 5.7% in Portugal to 21.3% in France, Several member states showed increased odds of ever e-cigarette use such as Malta (aOR = 5.46), Reports of trying e-cigarettes in 2014 were more common than in 2012 (aOR = 1.90), Highest prevalence of having ever tried e-cigarettes was found among: - current smokers: aOR = 23.36 compared to never-smokers - former smokers: aOR = 6.54 compared to never-smokers - younger people aged 18–24: aOR = 5.75 compared to those aged 55 years and older - those living in urban areas: aOR = 1.21 compared to those living in rural areas, - respondents who completed their education at age 20 or older: aOR = 1.65 compared to those who completed education at age 15 or younger, Proportion of current users among ever-users 2014: 15.3%, varying from 1.7% in Slovenia to 28.9% in Portugal (Austria: 14.7%), Those who tried an e-cigarette to quit smoking had the highest likelihood to be current users (aOR = 2.82),
**Brozek et al. 2019** [25]	Survey performed between 2017 and 2018, as a part of the international multi-center cross-sectional study, Young People E-Smoking Study (YUPESS)	Belarus, Lithuania, Poland, Russia and Slovakia	14,352 university students aged 18–34 years	Overall e-cigarette ever-use: 43.7%, Highest prevalence of ever-use was found among students in Lithuania (56.6%) and lowest prevalence among students in Russia (33.4%), Compared to Belarus, students in Lithuania (OR = 1.97) and Poland (OR = 1.44) were more likely to try using e-cigarettes, whereas students in Russia (OR = 0.79) were less likely to try using e-cigarettes, Mean age of e-cigarette usage initiation: 18.2 ± 2.2 years (approximately 2 years later than initiation of conventional smoking), Current e-cigarette use: 1.1%, Compared to Belarus, students in all other countries were more likely to currently use e-cigarettes, with odds ratios varying from 1.74 (Slovakia) to 2.37 (Lithuania), Current dual use of traditional cigarettes and e-cigarettes: 1.8%, Compared to Belarus, students in Poland were less likely (OR = 0.93) whereas students in all other countries were more likely to be dual users with odds ratios varying from 1.07 (Slovakia) to 1.7 (Russia), Females were less likely to be ever-users (OR = 0.62), current users (OR = 0.34) or dual users (OR = 0.33) compared to males,
**Eichler et al. 2016** [26]	2016 computer-assisted telephone interviews using a figure questionnaire	Germany	4002 randomly-chosen persons, aged 14 and older	Overall prevalence of e-cigarette ever-use: 11.8% of which 70% had only tried out e-cigarettes, Prevalence of ever-use among: - smokers: 32.7%; never-smokers: 2.3% - ex-smokers who had quit smoking after 2010: 24.5%; ex-smokers who had quit smoking before 2010: 1.8% - men: 15%; women: 8%, Overall prevalence of current regular use of e-cigarettes: 1.4%, Current regular use among: - current smokers: 4.3%, never-smokers: 0.1%, ex-smokers who had quit smoking after 2010: 5.6%, Overall Prevalence of former regular use: 2.2%, Former regular use among: - current smokers: 6.0%, never-smokers: 0.3%, ex-smokers who had quit smoking after 2010: 8.3%, Special groups: - Age group 20–39: most frequently represented group (2.4% current regular use, 4.4% former regular use and 16.1% have-tried), - Blue collar workers: above average use (4.6% for current regular use, 6.4% for former regular use and 14.2% for have-tried), - School students: above average in terms of “have-tried” (10.4%) and “former regular use” (5.9%) but low regarding “current use” (0.7%), The most common frequency of e-cigarette use: among current and former users as well as smokers, smokers who quit in 2010 or later and those aged 20–59,
**Andler et al. 2016** [27]	2014 Health Barometer Survey	France	Representative random sample of 15635 individuals of the French population aged 15–75	Prevalence of e-cigarette ever-use: 25.7%, Among these: 23.4% were current vapers, 60% of daily and 47.6% of occasional cigarette smokers reported lifetime use of e-cigarettes, whereas only 12.3% of ex-smokers and 5.6% of never-smokers did so, Prevalence of lifetime e-cigarette use was highest at age 15–24 years (48.8% of males and 40.7% of females), E-cigarette lifetime-use was lower in females (22.7%) compared to males (28.7%), Prevalence of e-cigarette current use: 6%, Among these: about half were daily vapers (75% of daily users were smokers and 23.1% were former smokers), - 83.1% were smokers, 74.7% were daily smokers and 15% were former smokers (more than 98% of current e-cigarette users were or had been conventional cigarette smokers), - farm workers, craftsmen, retailers, and business owners who smoked conventional cigarettes were less likely to use e-cigarettes, compared to those with lower incomes and the unemployed, Prevalence of current e-cigarette use was highest at age 25–34 years (9.6% of males and 6.6% of females), E-cigarette current use was lower among females (5.2%) compared to males (6.8%), Prevalence of daily vapers: 2.9%, Fewer females (2.3%) than males (3.5%) were daily vapers, Prevalence of vaping ex-smokers: 0.9%, Average duration of e-cigarette use: 4 months,
**Kilibarda et al. 2017** [28]	2014 National Survey on the Lifestyle of Citizens of Serbia	Serbia	A representative sample of 5385 Serbians aged 18–64 years	Total prevalence of e-cigarette ever-use was found to be 9.6%, while for current e-cigarette use, it was 2%, Prevalence for current e-cigarette use was highest among former daily (3.9%) and current tobacco smokers (3.3%–3.4%), Sex, age, residential area, occupation, and smoking status were significantly associated with ever-use of e-cigarettes, while education was not Highest prevalence of e-cigarette ever-use was stated among: - females (9.7%): they were 25% more likely than men to be ever e-cigarette smokers - those aged 25–34 (13.7%) - those living in urban areas (11.4%), they were 53% more likely than those living in rural areas, Prevalence of lifetime e-cigarette use was highest among current daily smokers (20.2%), with smokers generally being three times more likely than non-smokers to have ever used e-cigarettes, It was found that sex, age, occupation, and current smoking status were significantly associated with current e-cigarette use. Highest prevalence was stated among: - females (2.3%) compared to men (1.6%), those aged 25–44 (3%; 95% CI 2.0–4.0) compared to those aged 55–64 (1.1%; 95% CI 0.6–1.7) - intellectual (3.4%), compared to businessman (2.7%), non-active (1.7%), and student (0.9%), current smokers (3.3%–3.4%),
**Ruokolainen et al. 2017** [29]	2014 population-based drug survey	Finland	3485 respondents out of a representative random sample (N = 7000) of Finns aged 15–69	Overall prevalence of e-cigarette ever-use: 12%; Ever e-cigarette use was significantly associated with - daily tobacco use (OR = 19.6) current snus users (OR = 14.2) - age 15–24 years (OR = 10) age 25–34 (O R= 6.6) age 35–44 (OR = 2.2) - male (OR = 2.1) females (OR = 1) - being a student (OR = 2.8 ) unemployed (OR = 1.7) - 50% of ever-users stated always using nicotine-containing e-liquids - 40% of daily smokers and half of those currently using snus had at least tried e-cigarettes - 16% of ever-users reported not knowing whether the e-cigarette they had used contained nicotine or not, Overall prevalence of e-cigarette current use: 2%; Current e-cigarette was significantly associated with - daily tobacco use (OR = 60.6) occasional (OR = 37.8) current snus users (OR = 12.6) - age 15–24 years (OR = 12) age 25–34 (OR = 11.6) age 35–44 (OR = 5.1) - male (OR = 2.3) compared to females (OR = 1) - being unemployed a student (OR = 3.9) student (OR = 2) - 77% of current users stated always using nicotine-containing e-liquids, 14% sometimes and 9% never using liquids, respectively, Daily or almost daily use was most common among current snus users (4.3%) and the unemployed (2.7%), Concerning current e-cigarette users, the smallest percentage of respondents that stated always using nicotine was found among 15–24-year-olds (47%) and the highest among 25–34-year-olds (65%),
**Gallus et al. 2014** [30]	2013 survey on smoking	Italy	3000 individuals aged ≥15 years, representative for the general Italian population aged 15 years and over.	Prevalence of ever-use: 6.8%, E-cigarette ever-use was inversely related to age, the ORs compared to participants aged 15–24 years being 0.56 for 25–44, 0.49 for 45–64, and 0.16 for ≥65 years, Participants with an intermediate level of education were more likely to have ever used e-cigarettes (prevalence 10.2%, OR = 1.91), compared to those with the education levels was low or high, In terms of smoking status, current smokers showed the highest prevalence of ever-use of e-cigarettes (20.4%) compared to ex-smokers (7%) and never-smokers (2.6%), Prevalence current use: 1.2%, Regular use among men (1.5%) and women (0.9%), Regular use among different levels of education was comparable to each other (low 1.0%, intermediate 1.6%, high 0.5%), According to age, current use of e-cigarettes was most frequent in those aged 15–24 (2.4%), continuously decreasing with increasing age until 0.3% among those aged ≥ 65, In terms of smoking status, current smokers showed the highest prevalence for current use of e-cigarettes (3.7%) compared to ex-smokers (2.8%) and never-smokers (0.1%), Among current users, 95.5% used e-cigarettes with nicotine and the remaining 4.5% used e-cigarettes with vapor and flavors only, Frequency of use among current users varied from 1 to 70 e-cigarettes per day (mean: 10 overall),
**Kock et al. 2019** [31]	Monthly repeat household survey between January 2014 and December 2017 (Smoking Toolkit Study)	England	The Smoking Toolkit Study involved 1700–1800 adults aged 16+ living in households in England	Current e-cigarette use among all adults: 5.5%, - respondents from the lowest of 5 social grades were twice as likely to use e-cigarettes compared with those from the highest grade, - past-year smokers: 21.3%, - respondents from the 3 lowest social grades had significantly lower odds of e-cigarette use compared with those from the highest grade, - smokers during a quit attempt: 34.6%, - no significant associations across the overall period between social grades and prevalence of e-cigarette use among smokers attempting to quit, - long term ex-smokers: 5.9%, - respondents from the second and third lowest social grades were twice as likely to use e-cigarettes compared with respondents from the highest grade, - the trend of ex-smokers using e-cigarettes increased from 2014 to 2017 across all social grades,
**Jawad et al. 2015** [32]	Survey in public street settings conducted between March 2013 and March 2014	Southeast London (England)	1176 adults of any age in six southeast, ethnically diverse London boroughs	Prevalence of ever e-cigarette use: 7.4%, Among these: 47.1% were currently non-cigarette users, E-cigarette use was significantly associated with younger age groups: those aged 18–24 used more e-cigarettes (14.2%) than those aged 55 years and over (1.3%), - non-white ethnicities: 14.9%, compared to 5.6% among those of white ethnicity, - use of waterpipe tobacco: 20.6% of waterpipe tobacco smokers used e-cigarettes compared to 1.5% of non-waterpipe tobacco smokers,
**Martinez-Sanchez et al. 2014** [33]	Survey conducted between May 2013 and February 2014 in the course of the longitudinal study, The Determinants of Cotinine phase 3 project	Barcelona (Spain)	A representative sample of the adult (≥16 years old) population of Barcelona (n = 736)	Prevalence of e-cigarette ever-use: 6.5%, Among these, 75% were smokers, 22.9% were former smokers and 2.1% were never-smokers, 62.5% of ever-users, used e-cigarettes containing nicotine, Highest prevalence of ever-use was found among: - men: 8% compared to women (5.3%, OR = 0.69) - younger people ≤ 44 years old - people with intermediate education level: 9.8% (OR = 1.42 compared to low education level), - current smokers: 21.1% (OR = 54.57 compared to never-smokers) - current smokers with a high cigarette dependence score: 46.4% (OR = 3.96 compared to low-medium cigarette dependence score), Overall 1.6% were current e-cigarette users, 2.2% past users and 2.7% had only experimented with e-cigarettes,
**Goniewicz et al. 2012** [34]	A survey among high school and university students conducted between September 2010 and June 2011	Poland	20240 students enrolled at 176 nationally-representative Polish high schools and universities, aged 15–24 years, of which 13250 responded to questions about e-cigarettes	Prevalence of e-cigarette ever-use: among all students aged 15–24: 20.9%, among high school students aged 15–19: 23.5%, among university students aged 20–24: 19.0%, among never-smoking students: 3.2%, Prevalence of e-cigarette use within the previous 30 days: - among all students aged 15–24: 6.9%, among high school students aged 15–19: 8.2%, among university students aged 20–24: 5.9%, Associated with ever-use of e-cigarettes in terms of predicted probabilities was: - ever-use of cigarettes: 38% vs. 8.8% of those who had never smoked; gender: male 26.9% vs. 13.9% of girls, - having a parent or partner who smokes: 23.6% vs. 16.1% of those without smoking parents or partners, - living in an urban area, Associated with current use of e-cigarettes in terms of predicted probabilities was: - current smoking: 11.3% vs. 0.8% of those not currently smoking, - having a parent or partner who smokes: 6.8% vs. 1.5% of those without smoking parents or partners, Highest risk of ever (> 50%) and current e-cigarette use (> 25%) was identified for cigarette smoking boys, regardless of their age and place of living and who had a parent or partner who also smoked, Lowest risk of ever (< 4%) and current e-cigarette use (< 1%) was identified for nonsmoking girls, regardless of their age, who lived in rural areas and had nonsmoking parents and partners,
**Kaleta****et al. 2016** [35]	The survey adapted from the Global Youth Tobacco Survey was conducted between November 2014 and May 2015	Piotrkowski District (Poland)	3552 secondary and high school students aged 13–19 years from Piotrkowski District (2645 secondary school students and 907 high school students)	Prevalence of e-cigarette ever-use among all participants: 22%, - never tobacco users: 10%, ever tobacco users: 33% (OR = 6.7 compared to never tobacco smokers), - current tobacco users: 26% (OR = 9.8 compared to never tobacco smokers), - boys: 22% (OR = 1.3) vs. 22% among girls, - those whose parents smoked tobacco: 25% (OR = 1.4) vs. 19% among those whose parents were non-smokers, - those most or all of whose friends smoked tobacco: 27% (OR = 2.3) vs. 13% among those none of whose friends were smoking, - those who indicated alcohol consumption: 5% among moderate alcohol consumers–32% among binge drinkers vs. 9% among non-drinkers (OR = 5.3), - those who indicated that e-cigarettes were less harmful than current cigarettes: 21% (OR = 1.8) vs. 19% among those who indicated no difference in harmful effects between e-cigarettes and current cigarettes, - those who indicated that e-cigarettes were more harmful than current cigarettes: 44% (OR = 2.7) vs. 19% among those who indicated no difference in harmful effects between e-cigarettes and current cigarettes, - those whose mothers had the highest education level: 16% (OR = 0.5) vs. 24% among those whose mothers had the lowest education level, - those whose fathers had medium education level: 23% (OR = 1.5) vs. 21% among those whose fathers had the lowest education level, Prevalence of current e-cigarette use in the past month among all participants: 27%, - never tobacco users: 6%, ever tobacco users: 27% (OR = 7.5 compared to never tobacco smokers), - current tobacco users: 58% (OR = 32.5 compared to never tobacco smokers), - boys: 32% (OR = 1.7) vs. 21% among girls, - those whose parents smoked tobacco: 33% (OR = 1.4) vs. 23% among those whose parents were non-smokers, - those whose most or all of whose friends smoked tobacco: 48% (OR = 4.5) vs. 13% among those none of whose friends were smoking, - those who indicated alcohol consumption: 5% among moderate alcohol consumers – 41% among binge drinkers vs. 12% among non-drinkers (OR = 4.3), - those who indicated that e-cigarettes were less harmful than current cigarettes: 28% (OR = 2.1) vs. 29% among those who indicated no difference in harmful effects between e-cigarettes and current cigarettes, - those who indicated that e-cigarettes were more harmful than current cigarettes: 15% (OR = 0.3) vs. 29% among those who indicated no difference in harmful effects between e-cigarettes and current cigarettes, - those whose mothers had the highest education level: 13% (OR = 0.5) vs. 32% among those whose mothers had the lowest education level, - those whose fathers had the highest education level: 17% (OR = 0.6) vs. 33% among those whose fathers had the lowest education level, Predictors of continued e-cigarette use were: - male gender: OR = 1.4 compared to female gender - current tobacco smoking: OR = 3.0 compared never tobacco smoking - lack of knowledge about a ban on smoking in the school: OR = 1.4 compared to a ban on smoking in the school, Factors that protected from current e-cigarette use were: - higher parental education: OR = 0.5 compared to a low parental education, - perception of e-cigarettes as more harmful than tobacco cigarettes: OR = 0.2 compared to the perception of e-cigarettes being as harmful as tobacco cigarettes,
**Moore et al. 2015** [36]	Two data sets:2014 Child Exposure to Tobacco Smoke (CHETS) survey (‘CHETS Wales 2’) and 2014 Welsh Health Behaviour in School-aged Children (HBSC) Survey (‘HBSC Wales’)	Wales	CHETS Wales 2: 1601 school children in Year 6 (aged 10–11) within a nationally representative sample of 75 primary schools and HBSC: 9055 school students aged 11–16 in a nationally representative sample of 82 secondary schools	Prevalence of e-cigarette ever-use in year 6 primary school children: 5.8%, Among these, 3.7% reported using them just once, E-cigarette use was more prevalent among boys (7.2% vs. 4.6%), Secondary school students: 12.3%; Prevalence of e-cigarette ever-use among ‘never-smokers’: 5.3% at age 10–11 (Year 6), dropping to 2% in Year 7, before rising throughout secondary school to 8% by age 15–16 (Year 11), Most children in school years 6, 7 and 8 who reported ever-use of e-cigarettes have never smoked tobacco. Approximately half of Year 9 students have tried tobacco and among students in school years 10–11, a major proportion has tried tobacco, 42.8% of those who had used e-cigarettes on a few occasions stated that they had never smoked tobacco, Almost half of those who had tried smoking have tried an e-cigarette compared to only 4.8% of those who have never tried tobacco, The percentage of e-cigarette ever-users reporting to be current smokers increased from 10% at age 10–11 to 40% by age 15–16, Compared to never-smokers, the odds for e-cigarette use were more than 16 times greater for children aged 10–11 years old who had ever smoked tobacco and more than 17 times greater for current smokers, Prevalence of current e-cigarette use in secondary school students: 1.5%, Regular use was more likely among those who had smoked tobacco: 80% of current e-cigarette users reported having also smoked tobacco (RR= 66.30) but 72.1% of young people who had used an e-cigarette a few times and 43.2% of current e-cigarette users were not current smokers, Associations with current e-cigarette use were found for smoking weekly: RR = 121.15, smoking daily: RR = 115.38, lifetime cannabis use: RR = 53.03,
**Kinnunen et al. 2015** [37]	2013 nationwide Adolescent Health and Lifestyle Survey	Finland	A nationally representative sample of 9398 individuals aged 12, 14, 16 and 18 years, of which 3535 responded to the questionnaire	Prevalence of e-cigarette ever-use: 17.4%; Among these: - 8.3% were never-smokers - 12.6% had experimented only once or twice - 2% had used e-cigarettes more than 20 times - 65.7% stated use of nicotine e-liquids (among these, 2.9% were never-smokers), 23.5% used liquids without nicotine and 10.9% did not know whether the liquid had contained nicotine or not, Highest prevalence of e-cigarette ever-use was found among: - older adolescents for both sexes: 18.4% of 18-year-old girls had tried e-cigarettes once or twice compared to 0.3% of 12-year-old girls, - boys: 20.5% of 18-year-old boys had tried e-cigarettes once or twice compared to 18.4% of 18-year-old girls, Factors that protected from e-cigarette use were: - parents´ high level of education; being in employment; intact family, Associated with e-cigarette experimentation was: - daily smoking: OR = 41.35 compared to never smoking, - snus use: OR = 2.96 compared to never snus use, - waterpipe use: OR =2.21 compared to never waterpipe use, - children´s vocational education: OR = 2.06 compared to general education, - poor school performance: OR = 1.92 compared to school performance much or slightly better than class average,
**Treur et al. 2018** [38]	The survey among cohort I was conducted in 2014–2015 and the survey among cohort II was conducted in 2016–2017	Netherlands	Cohort I: 6819 adolescents from 19 secondary schools across the Netherlands, aged 11–17 years; Cohort II: 2758 adolescents from 14 educational institutes in the Netherlands, aged 14–21 years	Prevalence of ever-use of - e-cigarettes with nicotine: 13.7% (Cohort I), 12.3% (Cohort II) - e-cigarettes without nicotine: 29.4% (Cohort I), 27.6% (Cohort II) In the group of current users the highest frequency of use was detected among those who used nicotine-containing e-cigarettes (11.1 in Cohort and 9.3 in Cohort II)) compared to those using e-cigarettes without nicotine (7.9 in Cohort I and 4.8 in Cohort II), Compared to never using cigarettes, ever having used a conventional cigarette was associated with ever-use of e-cigarettes with nicotine: OR = 20.04 (Cohort I), OR = 19.70 (Cohort II), Compared to never using cigarettes, ever having used a conventional cigarette was associated with ever-use of e-cigarettes without nicotine: OR = 13.17 (Cohort I), OR = 7.31 (Cohort II), Highest prevalence of ever-use was found among: - those aged 16–17: OR = 1.9 (Cohort I) for ever-use of e-cigarettes with nicotine compared to 11–13-year-olds, E-cigarette ever-use was less likely among girls compared to boys; - OR = 0.52 (Cohort I), OR = 0.65 (Cohort II) for ever-use of e-cigarettes with nicotine, - OR = 0.51 (Cohort I), OR = 0.53 (Cohort II) for ever-use of e-cigarettes without nicotine,
**Dautzenberg et al. 2015** [39]	2013 repeated school-based survey	Paris (France)	A randomly selected, representative sample of 2% of schoolchildren (n = 3279) of the city of Paris aged 12–19 years	2013 prevalence of having experienced e-cigarettes among respondents: 17.9% (boys: 19%, girls: 16.8%) compared to 9.8% in 2012, Experimentation rate increased with advancing age from 5% (12-year-olds) to 30% (16-year-olds), E-cigarette experimentation was highest among daily smokers (63%), followed by occasional smokers (38.7%), former smokers (37.3%) and non-smokers (7.1%), E-cigarette experimentation was significantly associated with - age: OR = 0.66 for participants older than 15 years compared to those age 15 or younger, - having ever smoked a cigarette: OR = 4.46 compared to never smoking a cigarette, - smoking less than 10 cigarettes: OR = 2.28 compared to non-smoking, - smoking 10 cigarettes or more: OR = 5.67 compared to non-smoking, - best friends being smokers: OR = 1.54 compared to the best friend being non- or ex-smoker, - siblings being smokers: OR = 1.88 compared to not having siblings, siblings being non- or ex-smokers, - experimentation of shisha: OR = 2.60 compared to never using a shisha, - cannabis use: OR = 1.90 compared to having never used cannabis, - having one parent who forbids smoking: OR = 1.84 compared to “No prohibition”, - having two parents who forbid smoking: OR = 2.32 compared to “No prohibition”, - the kind of school: OR = 0.66 for children attending private schools compared to children attending public schools, Prevalence of use in the last 30 days: 5.6%, Prevalence of use in the last 30 days among e-cigarette experimenters: 32.5%,
**Rennie et al. 2016** [40]	Survey conducted in Winter of 2014–2015	Hauts-de-Seine region (France)	1486 participants in their first year of “lycée”, aged 16 years, of which 1478 answered questions concerning e-cigarettes	Prevalence of e-cigarette experimentation: 54%, Among these, 20% had never tried standard cigarettes, Experimentation with e-cigarettes was predicted by - higher age: OR = 1.30 (compared to younger participants in terms of above and below median age), - higher socioeconomic status: OR = 1.10 (compared to lower socioeconomic status), - maternal smoking of standard cigarettes: OR = 1.63 (compared to maternal nonsmoking), - paternal smoking of standard cigarettes: OR = 1.93 (compared to paternal nonsmoking), - male gender: OR = 1.21 (compared to female gender),
**Babineau et al. 2015** [41]	2014 survey on e-cigarette use, tobacco use, and socio- demographic items	Ireland	A representative sample of 821 young people from 16 secondary schools in their fifth year of secondary school, aged 16–17	Prevalence of e-cigarette ever-use: 24.0%, E-cigarette ever-use was found to be more likely among: - young men: 26.5% compared to young women (21.5%, OR = 0.73) - students in designated disadvantaged schools: 32.6% (OR = 1.77) compared to students in non-disadvantaged schools (20.6%) - current tobacco smokers: 69.5% (OR = 54.85 compared to non-smokers) compared to ever-smokers (30.4%, OR = 11.20 compared to non-smokers) and never-smokers (4.2%) - young people born in Eastern Europe: 44.4% compared to 21.9% among those born in Ireland and 23.6% to those born elsewhere, Prevalence of current e-cigarette use: 3.2%, Prevalence of current e-cigarette use among current smokers: 10.6%, On-going e-cigarette use was just predicted by male gender with females being less likely than males to use e-cigarettes on a current basis (OR = 0.38),
**Geidne et al. 2016** [42]	2014 survey as part of a study on “School as a setting for ANDT (Alcohol, Narcotics, Doping, Tobacco) prevention”	Sweden	665 participants from four municipalities in compulsory school, grade 9 (15–16-year-olds)	Prevalence of e-cigarette ever-use was 26%; - e-cigarette with nicotine was 13%; without nicotine was 10%, - without knowing whether they contained nicotine was 6%, Identified risk factors for e-cigarette ever-use were: - smoking conventional cigarettes (OR = 14.6 compared to never smoking) having tried cigarettes (OR = 5.6 compared to never smoking), - having tried snus (OR = 2.2 compared to never using snus), - using or having used alcohol (OR = 4.4 compared to never using alcohol), - having smoked a water pipe (OR = 3.2 compared to never smoking a water pipe), - not living with both parents (OR = 2.2 compared to always living with both parents), - having older siblings (OR = 1.7–1.8 compared to not having siblings), - not enjoying school (OR = 2.7 compared to stating very good/good school satisfaction), Having many books at home indicated less use of e-cigarettes (OR = 0.5–0.6 compared to having few books at home), Participants from southern Sweden smoked e-cigarettes more (50% ever-use) than those in the other two municipalities (17% ever-use in western municipalities),
**Douptcheva et al. 2013** [43]	Analysis as part of the Cohort Study on Substance Use Risk Factors (C-SURF), with data collected between August 2010 and February 2013	Switzerland	5081 young Swiss men enrolled during mandatory visits at army recruitment centers	Use of e-cigarettes in the past 12 months among: - all participants: 4.9%; among these, 12.0% used them daily, - current smokers: 9.3%; among these, 12.2% used them daily, - former smokers: 1.6%; among these, 13.6% used them daily, - never-smokers: 0.4% (no daily use), E-cigarette use among current smokers was significantly associated with secondary education: OR = 1.5 compared to tertiary or primary education, German-speaking region: OR = 1.3 compared to French-speaking region,

Note: CI = confidence interval; OR = odds ratio; aOR = adjusted odds ratio; RR = relative risk ratio; bold indicates the prevalence of e-cigarette ever-use, current, current, or daily use

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
