# Peer review of "Use of Electronic Cigarettes in European Populations: A Narrative Review"

_ijerph, 2020, doi:10.3390/ijerph17061971_

Round 1
Reviewer 1 Report
Electronic cigarettes are gaining significant popularity globally in recent years. It is critical to conduct a literature review of e-cigarette use in European countries as they seem to perceive e-cigarette use differently from the US. The authors included published articles on the prevalence of and reasons for e-cigarette use between 2011-2019, and stratifications by determinants were helpful. I do have a few comments that might be conducive to further improve it.
Line 38: what are controversies surrounding the use of e-cigarettes? Long-term health effects are unclear? Heavy metals and so on? These have not been mentioned in the previous paragraph. Line 39-41: “According to …”. This sentence is very vague and without any citations to support. What direct hazard and what harmful substance? It seems that the authors touched on that in Line 56-62. Probably need to restructure the introduction by putting similar statements regarding the same topic together in one paragraph. The authors might also want to mention vaping product use-associated lung injury in the US in the paragraph of Line 56-62. https://www.lung.org/lung-health-and-diseases/lung-disease-lookup/evali/ How did the authors come up with the search strategy? ‘prevalence of electronic cigarettes use’ or ‘e-cigarettes use’ or ‘electronic nicotine delivery system’ or ‘vaping’ or ‘frequency of e-cigarette use’. To me, some obvious keyword combinations are missing, such as “electronic cigarette”, “e-cigarette use” (the authors used plural forms), “Vaper”. The way the authors separated populations of interests into “general population” and “young populations” (11-34 year old) is a bit confusing when I read 3.1 and 3.2 because of their overlap. Any justification for lumping 18-34 year old with adolescents together as “young populations”. I would suggest having two groups: adults (18 year old and above) vs. (teenagers and adolescents). Line 198: Reasons for e-cigarette experimentation among adolescents probably deserve a paragraph to describe. They could be different from adults. Line 235: “People form eastern European countries used e-cigarettes more often in comparison to the European regional average”. Would it be possible to dig it a bit deeper? Why is the pattern displayed? Culture, social norm, and/or regulation? Line 280 Limitations: This part is not about limitations of the review per se, rather than challenges and methodological issues in previous papers. The authors might want to frame it as future research directions. I thought it was about the flaws of this narrative review.Author Response
Thank you for giving us the opportunity to submit a revised draft of the manuscript “Use of electronic cigarettes in European populations - a narrative review” to the International Journal of Environmental Research and Public Health.
We appreciate the time and effort that the editorial staff and the reviewers have dedicated to providing your valuable feedback on the manuscript. We have been able to incorporate changes to reflect most of the suggestions provided by the reviewers, which have been highlighted in the manuscript. Attached is a point-by-point response to the reviewers’ comments and concerns.

Reviewer 2 Report
ms #: ijerph-713670 Use of electronic cigarettes in European populations - a narrative review; Kapan et al.
This manuscript describes a subjective review of electronic cigarette use in general and young populations in Europe. The data appear to have been collected between 2012 and 2018, with the year varying across samples. The authors reported finding that in general populations the prevalence of current electronic cigarette use ranged from 0.2% to 5.5%, ever use ranged from 5.5% to 25.7%, daily use from 1% to 2.9%. They reported that among young adult populations current use ranged from 1.1% to 27%, ever use from 5.8% to 56.6% and regular use was at 1.5%. They reported that use of combustible cigarettes was highest amount electronic cigarette users, ranging from 20.4% to 83.1%, followed by ex-smokers at 7% to 15%. They reported that electronic cigarette use was also more likely in men and youth.
This manuscript focuses on an important topic in term of the prevalence of electronic cigarette use in European populations. I have some comments and questions, which might help to clarify the manuscript and its conclusions.
1) In the introduction the authors described studies that suggested that electronic cigarettes are effective for smoking cessation. However, one of these studies was an observational study and causal conclusions are not justified, the clinical trial was inadequately powered to actually establish any significant differences, and the review of such studies also suggested that there was inadequate power to detect differences. One issue with the clinical trials (e.g., Bullen et al., 2013) include that cessation rates were curiously low and lower than expected in many other studies that have included nicotine patch. Can the authors please address the limitations of these studies?
2) In the introduction, the authors do not mention the public health risk associated with youth transitioning to using combustible cigarettes after using electronic cigarettes, although they began to speak to this issue in the discussion. Could the authors begin to address this in the introduction as well?
3) Importantly could the authors please clarify if they consider their general populations general populations or adult populations? Given that the ages in the “general“ populations described are as low as 14 or 15, perhaps it might be more appropriate to consider them general populations. Could they please clarify the title of table 1 in the appendix where in the parenthesis they have “teenager, adolescents and young adults, aged 11-34 years old “, but go on to describe the general population samples.
4) At one point in the discussion the authors state that the most frequent reason for youth starting to use electronic cigarettes was to stop or reduced tobacco consumption, however in other areas the authors state that the main reason for electronic cigarette use in youth was “curiosity “and that peer use also plays a big factor. Could they please clarify what their review suggests to be the biggest factor?
5) Can the authors show some of their results in a graph/figure?
6) Given the variation in years in which the data were collected across samples can the authors described the trends in use across time?
7) In the discussion, the authors begin to compare the results in European samples to U.S. samples, however it is unclear the exact numbers that they are comparing. Could the authors please clarify which European numbers they are comparing to U.S. numbers? Could the authors speak to why there may be variation across European and American samples due to cultural and/or policy differences?
Author Response
Thank you for giving us the opportunity to submit a revised draft of the manuscript “Use of electronic cigarettes in European populations - a narrative review” to the International Journal of Environmental Research and Public Health.
We appreciate the time and effort that the editorial staff and the reviewers have dedicated to providing your valuable feedback on the manuscript. We have been able to incorporate changes to reflect most of the suggestions provided by the reviewers, which have been highlighted in the manuscript. Attached is a point-by-point response to the reviewers’ comments and concerns.

Round 2
Reviewer 1 Report
After reading the revised version and response letter, I believe that the authors have addressed most of my comments and the manuscript has been significantly improved.
Author Response
We appreciate the time and effort that the reviewers have dedicated to providing your valuable feedback on the manuscript. We thank the Reviewer for their positive comment and careful review, which helped improve the manuscript. Please find our answer to comments as well as suggested text changes in green.